# Towards Scalable Explainable AI: Using Vision-Language Models to Interpret Vision Systems

## Abstract

Explainable AI (xAI) is increasingly important for the trustworthy deployment of vision models in domains such as medical imaging, autonomous driving, and safety-critical systems. However, while saliency maps provide useful information about vision models, current explainable AI (xAI) methods remain bottlenecked by manual inspection of saliency maps or explaining them sample by sample without aggregation to explain their behaviors on large datasets, which makes large-scale analysis time-consuming and subjective. To address this, we propose a scalable automated pipeline that leverages Vision-Language Models (VLMs) to evaluate saliency-based explanations at both sample and dataset levels. Our method uses masked CAM images and prompts VLMs to generate descriptions for each sample, score attention quality, and aggregates results into a confusion-matrix framework for systematic analysis. We validate the pipeline on COCO, ImageNet, and PASTA datasets to show the method's ability and reliability. The result shows that our pipeline achieves 0.78 Pearson correlation with human judgments, outperforming traditional metrics and other xAI methods in performance, usefulness, and human alignment. We also show that the framework enables practical applications such as detecting mislabeled/incorrect samples with a 0.893 F1-score on COCO and a 0.885 F1-score on ImageNet, demonstrating its utility for scalable model evaluation and data auditing.

## 1 Introduction

Understanding how vision models make decisions is crucial for building reliable and trustworthy AI systems, especially in safety-critical applications. While numerous explainable AI (xAI) methods such as Class Activation Mapping (CAM) (Zhou et al., 2016), GradCAM (Selvaraju et al., 2017), ScoreCAM (Wang et al., 2020b), LIME (Ribeiro et al., 2016), and TCAV (Kim et al., 2018) have been proposed, their practical use remains limited by a lack of automation. Most existing approaches explain model behavior at the instance level, requiring researchers to manually inspect saliency maps or generated explanations for large numbers of images. This process is time-consuming, labor-intensive, and often subjective, making it difficult to scale analyses to large datasets or to capture the general behavior of vision models despite the need to understand how they learn and function on large datasets. TCAV partially addresses this by working at the dataset level, but it depends heavily on manually curated concepts. Similarly, frameworks such as LangXAI (Nguyen et al., 2024) automate description generation with Vision-Language Models (VLMs), yet still require human effort to summarize results and identify trends across datasets.

To overcome these limitations, we propose a scalable and automated pipeline that integrates saliency-based methods with VLMs to explain and evaluate vision models both at the single-sample and dataset levels. At its core, the pipeline generates saliency maps, transforms them into masked images, and leverages VLMs to produce interpretable explanations. These outputs are automatically aggregated into a confusion-matrix-based framework, which summarizes model behavior across an entire dataset. By reducing the reliance on manual inspection, our evaluation pipeline enables efficient discovery of failure cases, identification of systematic biases, and analysis of attention patterns in a scalable way.

This work makes three key contributions:

1. We introduce a fully automated evaluation pipeline that combines saliency-based xAI methods with VLMs to explain vision models at scale.

2. We demonstrate the pipeline's usability by using it to analyze image classification and segmentation models, detecting their potential biases and attention trends.

3. We show that our pipeline can also be used in data analysis tasks, demonstrating it with experiments to filter mislabeled and incorrect image samples.

By automating the process of analyzing and summarizing model explanations, this research addresses one of the bottlenecks in xAI: the dependence on manual effort. Our pipeline helps move explainability beyond single-sample inspection toward scalable, dataset-wide analysis, an essential step for integrating xAI into vision model development and ensuring transparent, reliable AI systems.

## 2 Related Work

Although many frameworks focus on evaluating vision model performance with metrics like accuracy, IoU, ensuring transparency and interpretability through explainable AI (xAI) is also crucial (Gunning & Aha, 2019; Zhao et al., 2015). xAI includes a variety of techniques to make machine learning models more interpretable and is generally classified as model-agnostic and model-specific methods (Lundberg & Lee, 2017). Model-agnostic approaches, applicable to any model, often assess feature importance, while model-specific methods leverage internal model structures for explanation (Bach et al., 2015). For vision tasks, popular techniques such as LIME (Ribeiro et al., 2016), TCAV (Kim et al., 2018), and saliency-based methods, including CAM (Zhou et al., 2016), Grad-CAM (Selvaraju et al., 2017), Grad-CAM++(Chattopadhyay et al., 2017), LayerCAM (Jiang et al., 2021), ScoreCAM (Wang et al., 2020a), EigenCAM (Muhammad & Yeasin, 2020), and XGradCAM (Oquab et al., 2015; Wang et al., 2020b) highlight regions important for predictions (Itti et al., 1998; Kümmerer et al., 2014; Zhao et al., 2015). These tools are especially valuable in fields like healthcare (Borys et al., 2023; Kakogeorgiou & Karantzalos, 2021; Kim & Joe, 2022), although many still require expert interpretation, which poses challenges to integration into development workflows.

The development of Vision-Language Models (VLMs) expands the capabilities of LLMs such as Qwen (Bai et al., 2023), Llama (Touvron et al., 2023), PerspectiveNet (Nguyen, 2024), and Phi (Li et al., 2023b) by enabling them to process visual information and text simultaneously (Ranasinghe et al., 2024; Liu et al., 2023). VLMs use vision models such as CLIP (Radford et al., 2021) to excel in multimodal tasks. Prominent examples include Flamingo (Alayrac et al., 2022), BLIP (Li et al., 2022), which integrates a visual encoder with an LLM via a querying transformer (Li et al., 2023a), and different VLMs such as GPT-4o, Qwen-VL (Bai et al., 2023), and Llama Vision (Chu et al., 2024), show strong ability to understand visual data. Consequently, they are used in many applications, including evaluating vision models (Chen et al., 2024), and more recently, analyzing deep learning models. Some research applying VLMs to model analysis can be listed, such as LangXAI (Nguyen et al., 2024), which explored the potential of using VLMs to generate explanations for visual recognition based on the intensity of colors extracted from CAM methods, MAIA (Shaham et al., 2024), which combines VLMs with programming tools to perform experiments to create descriptions of how model affected by changes from inputs, and a human-LLM alignment study (De Bona et al., 2024) in xAI. However, one limitation of the above methods is the lack of scalability on large datasets and model comparison. In particular, LangXAI generates a description for each sample without summarizing information on a large dataset; while MAIA can only focus on a small set of images (the top-activating images) each time, and neither of them rates the model's ability in image understanding.

Some other research in XAI, including human-centered evaluation, assesses CAM quality through metrics or human studies: Quantus (Hedström et al., 2023) contains over 30 metrics for systematic evaluation; sanity checks (Kindermans et al., 2019) exposed saliency method unreliability; psychophysics and game-based frameworks (Colin et al., 2022; Morrison et al., 2023) connect explanations to human perception; and PASTA (Kazmierczak et al., 2024) unified these for automated perceptual scoring. These approaches either rely on metrics misaligned with human judgment or require costly studies that cannot scale. Concept-based semantic validation methods like CIProVa (Parola et al., 2026) use foundation models to extract

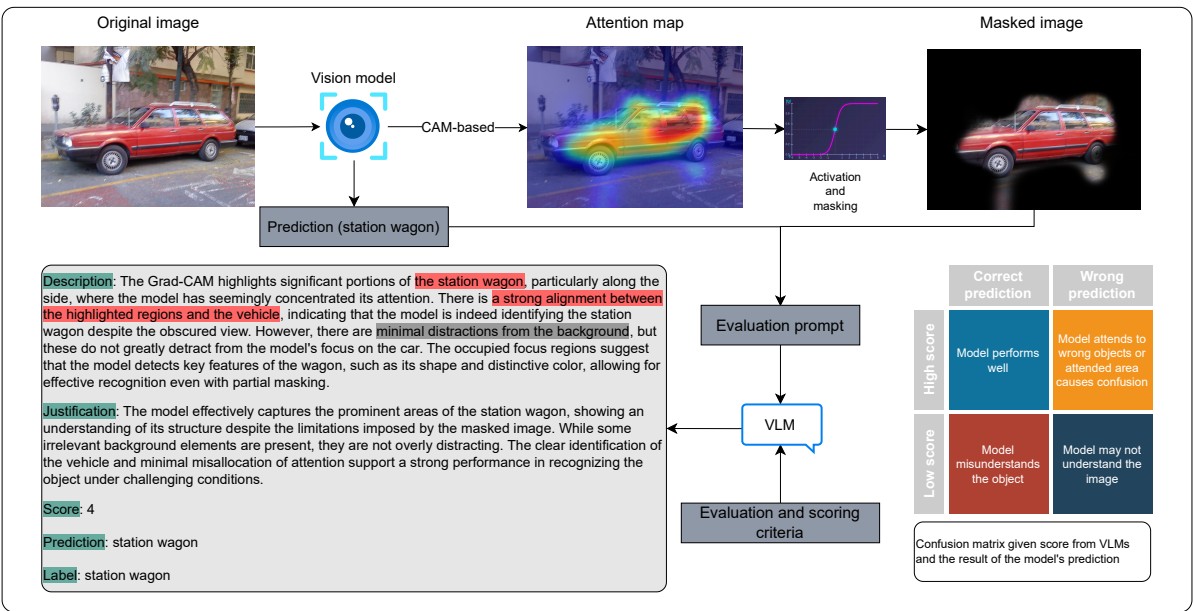

Figure 1: **The pipeline evaluates vision models' ability to understand an image.** The VLM model can describe, justify, and score the input image and the corresponding attention map. In the description, the model's interpretation of positive objects is highlighted in red, while gray illustrates the negative description.

visual concepts and validate saliency maps via Weight of Evidence. While CIProVa enables automated concept-alignment, it processes samples individually without quantitative scoring or cross-model comparison. Consequently, these evaluation streams remain disconnected: functional metrics sacrifice human-alignment, while concept-based methods lack systematic comparison mechanisms. To further bridge this gap, knowing that saliency images are useful to analyze deep learning models, we developed a scalable pipeline that uses VLMs to evaluate predictions from vision models, score them, provide explanations, and summarize the model's attention with a confusion matrix on a larger dataset. This method overcomes prior work by providing quantitative results on a larger dataset, thereby generalizing the use of xAI and better connecting training to understanding.

## 3 Methodology

We introduce a novel pipeline to explain vision models automatically. This pipeline combines CAM methods to visualize the model's attention and uses vision-language models to generate descriptions, evaluations, scores, and a confusion matrix. The entire proposed pipeline to explain and score vision models is illustrated in Figure 1.

### 3.1 Masked CAM image

The pipeline starts by feeding an image to vision models and getting a predicted result on the image. After that, different methods to extract models' attention, including CAM, LayerCAM, and more, are utilized to get an attention map of vision models on the image. Then, we apply a more general version of the *sigmoid* function to the attention map and get a mask for each image. The activation function is illustrated in Equation 1, where $v_{xy}$, ranging from 0 to 1, is the value of the attention map at position $(x, y)$, indicating the importance of the pixel, and $M_{xy}$ is the activated value at position $(x, y)$.

$$M_{xy} = \sigma(\alpha(v_{xy} - \beta)) = \frac{1}{1 + \exp\left(\alpha \cdot (\beta - v_{xy})\right)}. \tag{1}$$

In the equation, the values of $v_{xy} > \beta$ are scaled closer to 1 to highlight important regions, while $v_{xy} < \beta$ gradually decrease toward 0, reflecting reduced importance. Meanwhile, $\alpha$ controls the transition speed. The higher $\alpha$, the more sudden the transition from blacked-out to visible.

After achieving the mask, we apply it to the original image to hide regions with less attention according to the saliency-based method. This process is formulated in Equation 2, where we multiply each pixel in the original image $I$ by the corresponding value in the calculated mask $M$ in Equation 1 to achieve the final masked image $A$.

$$A_{xy} = I_{xy} \cdot M_{xy}. \tag{2}$$

The main reason we use the masked image instead of the heatmap overlay to explain the vision model's attention is to prevent image quality degradation, which can negatively affect VLM performance. A heatmap overlay can obscure important object features, thereby reducing a VLM's ability to interpret attention regions accurately. By blacking out areas outside the model's focus while retaining the attended regions, we preserve image quality for relevant objects while ensuring that VLMs focus exclusively on the regions required for evaluation. Furthermore, the vision model's attention should provide sufficient evidence to justify and explain its prediction. Insufficient evidence to recognize or distinguish objects within attended regions might suggest an underlying problem with the vision model.

## 3.2 VLM assessment

The result of the previous process is an image largely blacked out, except for areas the model considered important in its output. The masked image and the predicted label of the model are then fed to a VLM for evaluation and scoring. In the pipeline, VLMs are asked to find the relevance between the vision model's prediction and the visible object(s) in the masked image, and then explain further. Finally, VLMs score every pair of masked pictures and labels to quantify the model's ability. Our prompts for VLMs are provided in Section A.5.

## 3.3 Evaluation metrics

This section defines a confusion matrix for this pipeline based on the labels and generated explanation scores for each image. First, we select a threshold score to determine which generated scores indicate that the vision model struggles to understand the image. We then construct the matrix as shown in Figure 1, using the VLM scores and the model's prediction correctness for each sample. The proposed confusion matrix categorizes the model into four quadrants:

- **Correct:** The model attends to the correct object and predicts the correct label, indicating a strong understanding of the image.

- **Misunderstood object:** The model predicts the correct label, but its attention does not align with the target object, suggesting a misunderstanding of the object's visual appearance. When this pattern occurs frequently, inspecting the training data for this object category is recommended.

- **Attending to the wrong object:** The model focuses its attention on an object different from the labeled one and consequently makes an incorrect prediction. This behavior suggests difficulty in distinguishing the main object from contextual or background objects, and may be mitigated by architectures with stronger global attention mechanisms.

- **Lack of understanding:** The model fails to produce meaningful attention and predicts an incorrect label, indicating insufficient knowledge for the task. In such cases, training on a larger or more diverse dataset may be beneficial.

Given a large number of input samples, we can count the occurrences of each category and compute their proportions to obtain a comprehensive evaluation of the model. Furthermore, the resulting confusion matrix between attention alignment and prediction correctness can be used to derive actionable insights for mitigating the identified failure modes.

# 4 Experiment

We evaluated the pipeline's trustworthiness with four experiments to assess the VLMs' output (descriptions, scores), hyperparameter selection, and the usage of masked CAM and CAM images. The last one assesses our confusion matrix in predicting problems of trained vision models. The scoring system ranges from zero (random attention) to five (perfect attention), and saliency maps are extracted from the last layer. More details of why we chose this layer are reported in Section A.2.

## 4.1 Pipeline evaluation

**Pipeline scoring ability.** In the first experiment, we evaluated our pipeline by comparing VLM-based scores with human judgments and segmentation-based metrics. We sample 700 images from the COCO dataset for segmentation and extract saliency maps using ResNet18 with Grad-CAM. To avoid multi-object predictions, we filter for samples in which a single (or main) object occupies more than one-third of the image area and falls within the ResNet18 output classes, ensuring a dominant subject for analysis. Two authors independently rated the samples, achieving a Pearson correlation of 0.71. The annotating instruction is reported in Section A.3. Then, we average their ratings to obtain the human ground truth, alongside automatic metrics (IOU, Dice, PA, and F1), which is computed from the golden segmentation mask and attention mask (before applying to create the masked image), for comparison with the VLMs' scores using Pearson correlation. While human ratings may be high when the model attends to discriminative features or entire objects, metric scores only improve when attention aligns precisely with object regions at the pixel level. As shown in Table 1, when using masked CAM images, Gemini-2.5-flash-lite achieves correlations of 0.67 with human ratings, 0.31 with IOU, 0.31 with Dice, 0.33 with Pixel Accuracy (PA), and 0.31 with F1. Gemini-2.5-pro achieves the highest performance with 0.78 (human), 0.37 (IOU), 0.37 (Dice), 0.34 (PA), and 0.37 (F1). Using the original CAM images by modifying the LangXAI framework for scoring, Gemini-2.5-flash-lite obtains 0.64 (human), 0.28 (IOU), 0.29 (Dice), 0.29 (PA), and 0.29 (F1), while Gemini-2.5-pro achieves 0.75, 0.36, 0.36, 0.36, and 0.36, respectively, which is still lower than our method in most metrics. Compared to traditional XAI methods such as Delete and Insert (D&I) (Petsiuk et al., 2018), Average Drop (AD) (Chattopadhyay et al., 2017), and AOPC (Samek et al., 2015), which produce human correlations of 0.46, 0.34, and 0.11 and show consistently weaker alignment on segmentation metrics, our pipeline achieves substantially stronger consistency with both human judgments and pixel-level ground truth. Although direct comparisons are not strictly 'fair' in human correlation because these baselines were not designed with human interpretability in mind, these methods still have lower scores in metrics like F1, PA, IOU, and Dice based on the segmented areas, showing the potential of our approach. This also highlights a key limitation of traditional approaches in analyzing saliency images: the lack of contextual understanding. While these methods can identify attended regions as important, they cannot explain whether the attention truly aligns with the model's prediction, leading to lower-quality analysis and reducing the usefulness of information extracted from saliency maps. Finally, our approach benefits from the ability of vision-language models to reason about whether the model attends to the correct objects while maintaining better scores on different popular metrics. This leads to more reliable analysis if the model's attentions are biased, which most traditional methods fail to detect.

**Description and Justification.** Next, the authors checked the VLMs' output on 200 ImageNet samples to verify the quality of descriptions and justification for CAM and masked CAM images. In this experiment, they read the VLMs' output and decide whether those texts are acceptable. An output is unacceptable if the VLMs provide incorrect information, do not match the predicted object, or the score is not aligned with the justification and description. The results show that 85.58% of the GPT-4o-mini's generated samples on the masked CAM images are correct, while Gemini-1.5-flash achieves 79.41%. Meanwhile, results on the original CAM image show a lower rate; Gemini-1.5-flash achieves 54.22% and GPT-4o-mini achieves 75.62%. This indicates that weaker LLMs tend to benefit more from masked images.

**Hyperparameters.** The third experiment examines the impact of hyperparameters on framework correlation with humans and different metrics using a small VLM, *Qwen3-VL-2B-Instruct*. As reported in Table 2, the best human correlation is achieved with $\alpha = 25, \beta = 0.6$, reaching 0.703, while $\alpha = 15, \beta = 0.6$ achieves the highest pixel-level correlations across IOU (0.299), Dice (0.305), F1 (0.305), suggesting that a smaller

| Model | Method | Human | IOU | Dice | PA | F1 |
|---|---|---|---|---|---|---|
| Gemini Pro | Masked CAM | **0.78** | **0.37** | **0.37** | 0.34 | **0.37** |
|  | CAM image | 0.75 | 0.36 | 0.36 | **0.36** | 0.36 |
| Gemini Flash-lite | Masked CAM | 0.67 | 0.31 | 0.31 | 0.33 | 0.31 |
|  | CAM image | 0.64 | 0.28 | 0.29 | 0.29 | 0.29 |
| Gemma-3-27b | Masked CAM | 0.65 | 0.31 | 0.31 | 0.28 | 0.31 |
|  | CAM image | 0.51 | 0.22 | 0.23 | 0.26 | 0.23 |
| Baselines | D&I | 0.46 | 0.29 | 0.28 | 0.33 | 0.28 |
|  | Average Drop (AD) | 0.34 | 0.22 | 0.22 | 0.19 | 0.22 |
|  | AOPC | 0.11 | 0.21 | 0.21 | 0.11 | 0.21 |

Table 1: **Comparison across explanation methods on multiple evaluation metrics.** All values are Pearson correlations between explanation scores and reference metrics: human ratings, Intersection-over-Union (IOU), Dice, Pixel Accuracy (PA), and F1. We report results for Gemini Pro, Gemini Flash-lite, Gemma-3-27B, and model-agnostic baselines.

$\alpha$ with moderate $\beta$ better preserves spatial alignment with segmentation masks. In general, all hyperparameter configurations consistently outperform the original CAM used in the modified LangXAI method in both human and pixel-level evaluation, confirming the robustness of the proposed masking strategy. Notably, these results are obtained using a relatively lightweight VLM, showing that as the underlying model becomes weaker, the advantage of our pipeline over LangXAI becomes more pronounced.

|  | $\alpha = 15$ | | | $\alpha = 20$ | | | $\alpha = 25$ | | |
|---|---|---|---|---|---|---|---|---|---|
|  | $\beta = 0.4$ | $\beta = 0.6$ | $\beta = 0.8$ | $\beta = 0.4$ | $\beta = 0.6$ | $\beta = 0.8$ | $\beta = 0.4$ | $\beta = 0.6$ | $\beta = 0.8$ |
| Masked (Human) | 0.661 | 0.651 | 0.639 | 0.647 | 0.660 | 0.621 | 0.675 | **0.703** | 0.626 |
| Masked (IOU) | 0.278 | **0.299** | 0.275 | 0.286 | 0.273 | 0.262 | 0.261 | 0.264 | 0.280 |
| Masked (Dice) | 0.288 | **0.305** | 0.284 | 0.300 | 0.279 | 0.268 | 0.273 | 0.272 | 0.286 |
| Masked (PA) | 0.257 | 0.273 | 0.261 | **0.279** | 0.254 | 0.238 | 0.235 | 0.258 | 0.265 |
| Masked (F1) | 0.288 | **0.305** | 0.284 | 0.300 | 0.279 | 0.268 | 0.273 | 0.272 | 0.286 |
| Original (Human) |  |  |  |  | 0.272 |  |  |  |  |
| Original (IOU) |  |  |  |  | 0.134 |  |  |  |  |
| Original (Dice) |  |  |  |  | 0.132 |  |  |  |  |
| Original (PA) |  |  |  |  | 0.123 |  |  |  |  |
| Original (F1) |  |  |  |  | 0.132 |  |  |  |  |

Table 2: **Pearson correlation of Masked CAM (Ours) and Original CAM (LangXAI) with humans and pixel-level metrics across hyperparameters** $\alpha \in \{15, 20, 25\}$ **and** $\beta \in \{0.4, 0.6, 0.8\}$**.** Bold values indicate the highest correlation across all settings per row. We used *Qwen3-VL-2B-Instruct* for this experiment.

## 4.2 Evaluation with Human Alignment

We evaluate our method on the PASTA (Perceptual Assessment System for explanaTion of Artificial Intelligence) benchmark (Kazmierczak et al., 2024), a human-centric framework that addresses the limitation of conventional XAI metrics, such as faithfulness or sensitivity, which overlook how human users perceive explanations. PASTA collects human preference annotations over explanations generated on four diverse image datasets, and trains an automated scoring model to replicate these judgments at scale. Explanations are scored along six criteria: **Q1** understandability; **Q2** accuracy of the explanation with respect to the model's reasoning; **Q3** completeness; **Q4** accessibility across diverse demographics; **Q5** stability under minor input perturbations; and **Q6** sensitivity under strong perturbations that flip the prediction. In this experiment, we compare different xAI methods, including ours, with the PASTA-score using Pearson correlation (with a 0.501 Spearman correlation coefficient compared with humans). By indirectly comparing xAI scores with

humans using the PASTA-score, the higher Pearson correlation indicates that the method better aligns with human decisions. The experiment is conducted using *Qwen3-VL-2B-Instruct*, which is a very small language model compared to previous experiments.

As shown in Table 3, our method achieves the best mean score (0.2960) and ranks first on Q1 to Q4 and Q6, demonstrating strong alignment with human judgment across the core interpretability criteria like understandability, accuracy, completeness, accessibility, and sensitivity. On Q5, most methods yield negative correlations except the Delete and Insert method, indicating that none of the evaluated methods align well with human judgment on explanation stability. AOPC's consistently negative scores across all criteria confirm that model-centric perturbation metrics are poorly aligned with human perception. This experiment shows that our method, despite using a very small vision language model, can achieve good performance compared to other methods. Furthermore, the high gap between our method and LangXAI strengthens that our solution is better for a smaller vision language model in generating explanations and judgments.

| Method | Q1 | Q2 | Q3 | Q4 | Q5 | Q6 | Mean |
|---|---|---|---|---|---|---|---|
| **Ours** | **0.4123** | **0.4643** | **0.3643** | **0.3666** | $-0.1385$ | **0.3106** | **0.2960** |
| AvgDrop | 0.3632 | 0.3542 | 0.2519 | 0.3281 | $-0.2006$ | 0.3003 | 0.2329 |
| LangXAI | 0.2709 | 0.1898 | 0.1462 | 0.1867 | $-0.0605$ | 0.2521 | 0.1642 |
| Uncertainty | 0.1612 | 0.1285 | 0.1123 | 0.1552 | $-0.1707$ | 0.2180 | 0.1008 |
| DaI | 0.0359 | 0.0827 | 0.0174 | 0.0193 | **0.1205** | $-0.0513$ | 0.0374 |
| AOPC | $-0.2719$ | $-0.3220$ | $-0.2390$ | $-0.2700$ | $-0.0537$ | $-0.2548$ | $-0.2353$ |

Table 3: **PASTA benchmark results (Pearson correlation with PASTA-score).** Higher is better. Best in **bold**. *Qwen3-VL-2B-Instruct* is used for this experiment.

### 4.3 Usefulness Evaluation

We further evaluate on the COCO dataset, which contains diverse scenarios (e.g., occlusion, small objects), using the Utility metric (Colin et al., 2022). This metric computes the ratio of correct predictions made with explanations to those made without. An xAI method is beneficial when its utility score is larger than one. As human studies are costly and not scalable to COCO, we replace human evaluators with a vision-language model (VLM), following the original intent of measuring how well an interpreter can infer model predictions from explanations. We use *Qwen3-VL-2B-Instruct* as a proxy interpreter to enable large-scale and reproducible evaluation. Given that the task is constrained classification over a predefined label set, a lightweight VLM is sufficient.

In the experiment, we use ResNet50 as a main model to explain, and for each sample, we take the top-20 predictions of this model as candidate labels and ask the VLM to classify the original image and the corresponding explanations. This action ensures that the correct label for each image is not too obvious, and the audience (VLM) needs sufficient information from the explanation to predict correctly. For text explanations, we additionally report results with masked labels (all labels that appear in text are removed) to avoid leakage.

As shown in Table 4, our method achieves the best performance for both with and without labels in the text, outperforming LangXAI and all saliency-based methods. Next, by introducing labels in the description, VLM tends to make better predictions. This shows that any future xAI methods that produce explanation text must hide predicted labels of vision models for fair judgment. Lastly, this result shows that our xAI method produces more useful explanations than the compared methods.

### 4.4 The Impact of Prompt Style

In this section, we investigate how variations in prompts influence the performance of our pipeline, potentially yielding positive or negative effects. In this experiment, we maintain the same output requirements (description, justification, and score) and inputs (evaluation criteria, scoring criteria, and image description)

| Method | Description | Justification | Description (w/o label) | Justification (w/o label) |
|---|---|---|---|---|
| **Ours** | 1.706 | **2.014** | 1.400 | 1.407 |
| LangXAI | 1.447 | 1.542 | 1.324 | 1.400 |
| XGradCAM | | 1.241 | | |
| CAM | | 1.213 | | |
| GradCAM | | 1.210 | | |
| GradCAM++ | | 1.206 | | |
| LayerCAM | | 1.173 | | |

Table 4: Utility$_k$ on COCO. Values above 1 indicate that the explanation helps humans predict correctly more often than without it. Higher is better.

to ensure that we are comparing only prompting styles, without modifying the underlying target or pipeline functionality. Accordingly, we employed two additional distinct prompts using the Gemma-3-27B model. The first prompt is a concise version of our original that shortens the image description, evaluation criteria, and scoring criteria while maintaining their semantic meaning and the pipeline objectives. The second is an extended prompt that requires more detailed and longer image descriptions and justifications. The results are reported in Table 5. Overall, we observe that prompt variations are not significant, with the original prompt achieving the highest performance across all metrics.

| Prompt | Human | IOU | Dice | PA | F1 |
|---|---|---|---|---|---|
| Original | 0.651 | 0.316 | 0.317 | 0.286 | 0.317 |
| Extended | 0.640 | 0.306 | 0.306 | 0.280 | 0.306 |
| Concise | 0.651 | 0.301 | 0.301 | 0.273 | 0.301 |

Table 5: **Comparison of different prompt types across various metrics.** The evaluation is conducted on 700 samples from the COCO datasets.

### 4.5 Model analysis

| Model | CH | CL | WH | WL | Avg |
|---|---|---|---|---|---|
| **Segmentation Models** | | | | | |
| DeepLabv3-ResNet50 | 76.9 | 5.4 | 10.8 | 6.9 | 3.98 |
| DeepLabv3-ResNet101 | 80.9 | 4.9 | 7.8 | 6.4 | 3.94 |
| LRASPP-MobileNet v3-Large | 66.2 | 6.4 | 12.3 | 15.2 | 3.52 |
| FCN-ResNet50 | 71.1 | 5.9 | 14.7 | 8.3 | 3.73 |
| **Classification Models** | | | | | |
| ResNet18 | 64.2 | 4.9 | 17.6 | 13.2 | 3.45 |
| ConvNeXt-tiny | 66.2 | 28.4 | 2.9 | 2.5 | 3.00 |
| MaxViT-t | 67.1 | 23.0 | 6.9 | 2.9 | 3.14 |
| Efficientnet-b1 | 74.0 | 14.2 | 7.4 | 4.4 | 3.43 |

Table 6: **Confusion matrix-based attention analysis of different vision models. CH**, **CL**, **WH**, **WL** are referred to as Correct-High, Correct-Low, Wrong-High, Wrong-Low in the confusion matrix (percentage). Meanwhile, **Avg** denotes the Average Attention Score from the pipeline. We evaluate those models using PASCAL VOC (Everingham et al., 2025) and ImageNet.

We further experimented to analyze how different vision models rely on attention mechanisms when making predictions. Table 6 presents the distribution of samples across four categories: Correct-High (CH), where the model makes a correct prediction with high attention focused on the target object; Correct-Low (CL),

where the prediction is correct but attention is low or scattered; Wrong-High (WH), where the model predicts incorrectly despite attending strongly to an object (often a secondary or boundary object); and Wrong-Low (WL), where the model both predicts incorrectly and fails to attend to any object meaningfully. More details of the experiment's setup are reported in Section A.2.

**For segmentation models,** the majority of cases fall under CH, with DeepLabv3-ResNet101 (Chen et al., 2017) achieving the highest CH rate (80.9%), followed by DeepLabv3-ResNet50 (He et al., 2015) (76.9%). This suggests that segmentation architectures generally rely on their attention mechanism to segment the correct objects and make predictions, which aligns with their dense pixel-level supervision. However, there are many cases where all segmentation models failed with high attention score, 10.8% for DeepLabv3-ResNet50, 7.8% for DeepLabv3-ResNet101, 12.3% for LRASPP-MobileNet v3-Large (Howard et al., 2019), and 14.7% for FCN-ResNet50 (Shelhamer et al., 2014), which indicates that some segmentation models do not fully utilize their attention mechanism for segmenting objects or their attentions are still too complex to segment from. LRASPP-MobileNet v3-Large exhibits the weakest attention stability among segmentation models (66.2% CH, 15.2% WL), suggesting that this model frequently fails to leverage contextual cues necessary for accurate object localization and segmentation, resulting in significantly lower performance compared to other architectures.

**For classification models,** the distribution varies more significantly. ResNet18 achieves only 64.2% CH and exhibits a high WH rate (17.6%) and WL (13.2%) with a very low rate on CL, indicating the model strongly depend on the attention mechanism to make decision and its attention captures nearly entire objects, if the model fails to capture the object (low attention score), the model is likely to fail the task. This also suggests that the model is susceptible to distraction from background objects, leading to erroneous predictions that pose potential security risks. EfficientNet-b1 (Tan & Le, 2019) improves CH to 74.0%, but still shows a non-negligible WH (7.4%) and WL (4.4%). Furthermore, its CL score is relatively high, suggesting stronger representational power but partial bias on localized features of the object or possible flaws in its attention mechanism. ConvNeXt-tiny (Liu et al., 2022) and MaxViT-t (Tu et al., 2022) demonstrate an interesting trend: while they achieve relatively high CH (66.2% and 67.1%), their CL ratios are unusually high (28.4% and 23.0%, respectively). This suggests that these models can make accurate predictions even when attention is not clearly aligned with the main object, similar to Efficientnet-b1, possibly due to learning more abstract or global representations rather than relying strictly on localized attention. However, this also implies a potential bias toward partial features, where small discriminative patches suffice for classification, rather than full-object reasoning.

Overall, the results suggest that segmentation models maintain more consistent attention alignment with the target object, whereas classification models vary in their reliance on attention. Models such as ConvNeXt-tiny and MaxViT-t appear less dependent on object-centered attention, favoring global or distributed feature learning, while traditional CNNs like ResNet18 rely more heavily on attention consistency but remain vulnerable to boundary confusions. This experiment highlights the ability of our pipeline to reveal intrinsic differences in how vision models allocate and utilize attention for decision-making.

## 5 Applications in Data Analysis

In the next set of experiments, we evaluate the ability of different frameworks to analyze the internal mechanisms of vision models for detecting flawed samples on two widely used datasets: COCO (Lin et al., 2014) and ImageNet (Deng et al., 2009) in two scenarios, to detect noisy labeled images with multiple objects and detect wrong labeled images from an augmented dataset using a vision model. This experiment shows that our pipeline can potentially be applied to data analysis tasks, not just evaluating vision models.

### 5.1 Incorrect Sample Detection

**Incorrect sample and datasets.** Flawed samples in 700 COCO samples primarily come from incorrect or ambiguous annotations in complex, multi-object scenes: some images contain multiple valid labels, and a small portion are simply mislabeled. For our evaluation, one of the authors manually annotated these flawed instances by strictly applying these criteria, specifically identifying images with ambiguous boundaries, multi-

object conflicts, or clear misassignments against the original COCO labels to establish the ground truth. These issues can confuse a vision model or lead to inconsistent predictions.

**Setup.** Our framework detects such cases by exploiting the correlation matrix and focusing on "WH" cases, where the model attends to the correct region but produces predictions inconsistent with the given label, suggesting a labeling issue. This means for each sample in the dataset, if it has a high score from our pipeline but ResNet18 predicts the sample's label incorrectly, that sample is marked as 'incorrect'. We further compare the results of our method with other methods like confident learning (CL) (Northcutt et al., 2021), low-normalized margin (LNM) (Yuan et al., 2025), Low Self Confidence (LSC) (Northcutt et al., 2021), prune-by-noise-rate (PNR) (Northcutt et al., 2021), and LangXAI. For the baseline, we use ResNet18, which marks a sample correct if its prediction matches the image label and vice versa. For other xAI methods, ResNet18's signal will be used to make predictions. More details of the experiment's setup are reported in Section A.2.

**Results.** As shown in Table 7, our method achieves accuracy 0.829, precision 0.924, recall 0.864, and F1 0.893. Compared to baselines, ResNet18 shows very high precision (0.966) but extremely low recall (0.679), accepting too many invalid samples. On the other hand, confident learning, low-normalized margin, and Low Self Confidence provide high recall (0.981, 0.975, and 0.993) but at the cost of lower precision, retaining many flawed samples. Proposed filtering strategies such as prune-by-noise-rate and low-self-confidence perform the best overall, with F1-scores of 0.940 and 0.934, respectively. Although not specifically designed for data filtering, our method offers a competitive trade-off: higher purity than recall-heavy baselines, and less restrictive than conservative filters, making it robust for dataset auditing in challenging scenarios.

| Method | Acc. | Prec. | Recall | F1 |
|---|---|---|---|---|
| ResNet18 | 0.714 | **0.966** | 0.679 | 0.797 |
| LangXAI-m | 0.818 | 0.920 | 0.855 | 0.886 |
| CL | 0.864 | 0.871 | 0.981 | 0.922 |
| PNR | **0.898** | 0.915 | 0.967 | **0.940** |
| LSC | 0.884 | 0.881 | **0.993** | 0.934 |
| LNM | 0.855 | 0.866 | 0.975 | 0.917 |
| **Ours** | 0.829 | 0.924 | 0.864 | 0.893 |

Table 7: **Comparison of methods for detecting flawed samples in the COCO dataset. Bold** indicates the best result and underline the second best. Abbreviations: LXM = LangXAI (modified), CL = Confident Learning, PNR = Prune by Noise Rate, LSC = Low Self Confidence, LNM = Low-Normalized Margin.

## 5.2 Mislabeled Sample Detection

**Mislabeled samples and datasets.** In this experiment, we further simulate large-scale augmentation pipelines by introducing noisy labels from vision models themselves (i.e., automatic relabeling). This allows us to test whether methods can identify mislabeled samples introduced by model-driven noise rather than human annotators. Following this, we defined a sample as mislabeled if the classifier's label does not match the golden label. For the dataset, we randomly sample 518 ImageNet images and use ResNet18 to generate noisy labels (i.e., automatic relabeling that may conflict with ground truth).

**Setup.** From the extracted CAMs and internal model signals, each xAI method must detect whether ResNet18's prediction is accurate. The baseline in this experiment is ResNet18, which treats all predictions as correct, disregarding any confidence or attention cues. Furthermore, in this scenario, our pipeline can only recognize two cases "CH" and "CL" as the "fake golden label" from the classification model always matches ResNet18's prediction. Thus, our pipeline will consider a sample to be wrong if it belongs to the "CL" case and vice versa. More details of the experiment's setup are reported in Section A.2.

**Results.** Table 8 shows that our framework achieves the highest accuracy (0.828) and precision (0.843), along with strong recall (0.932) and F1-score (0.885). In contrast, CL and LNM perform poorly across all metrics. PNR and LSC outperform CL and LNM but still trail our framework in both accuracy and

precision. Interestingly, LangXAI-m achieves a competitive F1 (0.885), though it slightly lags behind our framework in accuracy and precision. These results highlight that our framework, though not explicitly designed as a filtering tool, is particularly effective at suppressing model-induced labeling errors, which are common in large-scale automatic pipelines, by leveraging its ability to evaluate and analyze the models' attention mechanisms.

| Method | Acc. | Prec. | Recall | F1 |
|---|---|---|---|---|
| ResNet18 | 0.713 | 0.713 | **1.000** | 0.833 |
| LangXAI-m | 0.826 | 0.835 | 0.943 | **0.885** |
| CL | 0.735 | 0.758 | 0.924 | 0.832 |
| PNR | 0.723 | 0.722 | 0.994 | 0.836 |
| LSC | 0.742 | 0.762 | 0.929 | 0.837 |
| LNM | 0.716 | 0.747 | 0.911 | 0.821 |
| **Ours** | **0.828** | **0.843** | 0.932 | **0.885** |

Table 8: **Comparison of methods for detecting flawed samples in the ImageNet dataset. Bold** indicates the best result and underline the second best. Abbreviations: LangXAI-m = LangXAI (modified), CL = Confident Learning, PNR = Prune by Noise Rate, LSC = Low Self-Confidence, LNM = Low-Normalized Margin. ResNet18 is used to create 'fake' labels for this dataset, while other methods have to find out which image and label pair from ResNet18 is incorrect.

## 6 Conclusion

This paper proposed a novel framework to integrate CAM visualizations with VLM to explain vision models. The pipeline can be easily integrated into the evaluation process to provide more details, including text-based explanations, scores, and a confusion matrix. This pipeline's specialty is that it can provide assessments for both the sample-level and dataset-level, providing more insights for researchers. The research also highlights the potential of the VLM-as-a-judge paradigm for large-scale evaluation, explanation, and analysis of deep learning models, where the need to interpret complex mechanisms across large datasets is rapidly growing.

### Limitations

Despite being scalable and helpful in detecting scenarios where the vision models behave incorrectly, the pipeline still contains some limitations, including the dependence on VLMs to generate a correct description with a suitable score for each sample. Furthermore, the pipeline only utilizes saliency-based methods to extract the attention regions, but not methods like finding the decision boundary and other xAI visualization techniques.

### Potential risk

The quality of the generated descriptions is highly dependent on the performance of the VLM, despite it has similar or better performance in our evaluation. Therefore, the pipeline should be used only as a supporting tool, with the researcher remaining the primary decision maker in the analysis.

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

# A Appendix

## A.1 Computational cost

As our pipeline relies on a VLM, the overall cost varies across providers due to differences in pricing schemes and tokenization. To ensure transparency and facilitate reproducibility, we report the average input and output statistics, which can be used to estimate monetary cost depending on the chosen provider and tokenizer. For image input, we convert all images to $224 \times 224$. With these statistics, each sample running with Gemini-2.5-flash-lite costs about 0.00016$

|        | Length (chars) | Words   | Sentences |
|--------|----------------|---------|-----------|
| Input  | 2199           | 459     | 18        |
| Output | 1603.638       | 300.863 | 13.428    |

Table 9: Average input/output statistics for cost estimation.

## A.2 Experiment Setup

**Saliency methods.** Following the established practice in saliency-based visualization literature (Selvaraju et al., 2017; Zhou et al., 2016; Jiang et al., 2021; Wang et al., 2020a), we extract activation maps from the final convolutional layer of vision models. This choice is motivated by a principled trade-off between high-level semantic information and spatial localization fidelity: deeper layers capture class-discriminative object parts with larger receptive fields (Selvaraju et al., 2017), whereas convolutional layers preserve the spatial structure necessary for localization. Next, section 4.1 employs three methods (GradCAM, CAM, LayerCAM) across ResNet18 and MaxViT-t to ensure diverse attention patterns. All other experiments use GradCAM exclusively for consistency.

**Score thresholding.** We set the threshold at 1 based on the VLM scoring rubric, where scores smaller than or equal to 1 denote completely failed attention and larger scores indicate at least partial object capture. This separates cases where the model fails to attend to the target object (low score) from cases with meaningful attention (high score).

## A.3 Annotation Guide

Annotators evaluate each sample using two visual inputs: the saliency map (heatmap overlaid on the original image) provides context, and the masked CAM image isolates regions the model actually attended to, preventing annotators from using external visual information outside the model's focus. The model's predicted label is also shown.

Annotators score attention quality from 0 to 5, which is the same criteria as VLMs:

**0–1** Attention is irrelevant or misses the object entirely.

**2–3** Partial recognition: captures small parts or includes irrelevant areas.

**4–5** Strong alignment: mostly correct (4) or perfect capture without distraction (5).

## A.4 Examples of Model's Evaluation

We present additional qualitative results of our benchmark to analyze the effectiveness of our method and evaluation metrics. The example shown in Figure 2 demonstrates how the model's attention can sometimes focus on irrelevant features, but does not lead to reduced interpretability.

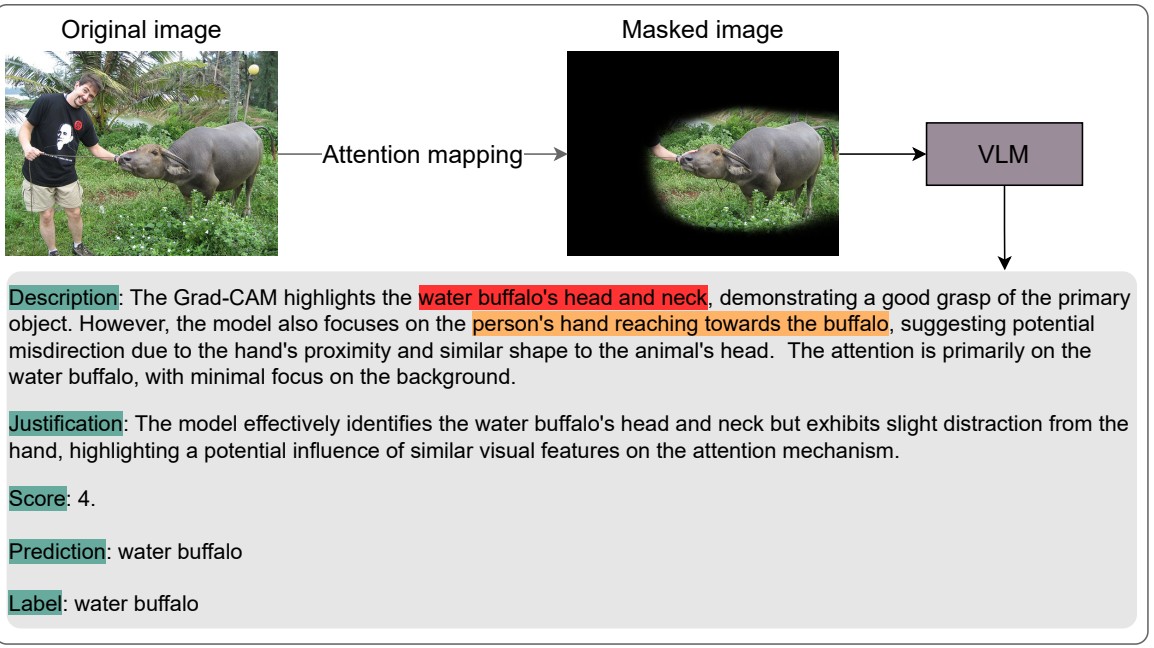

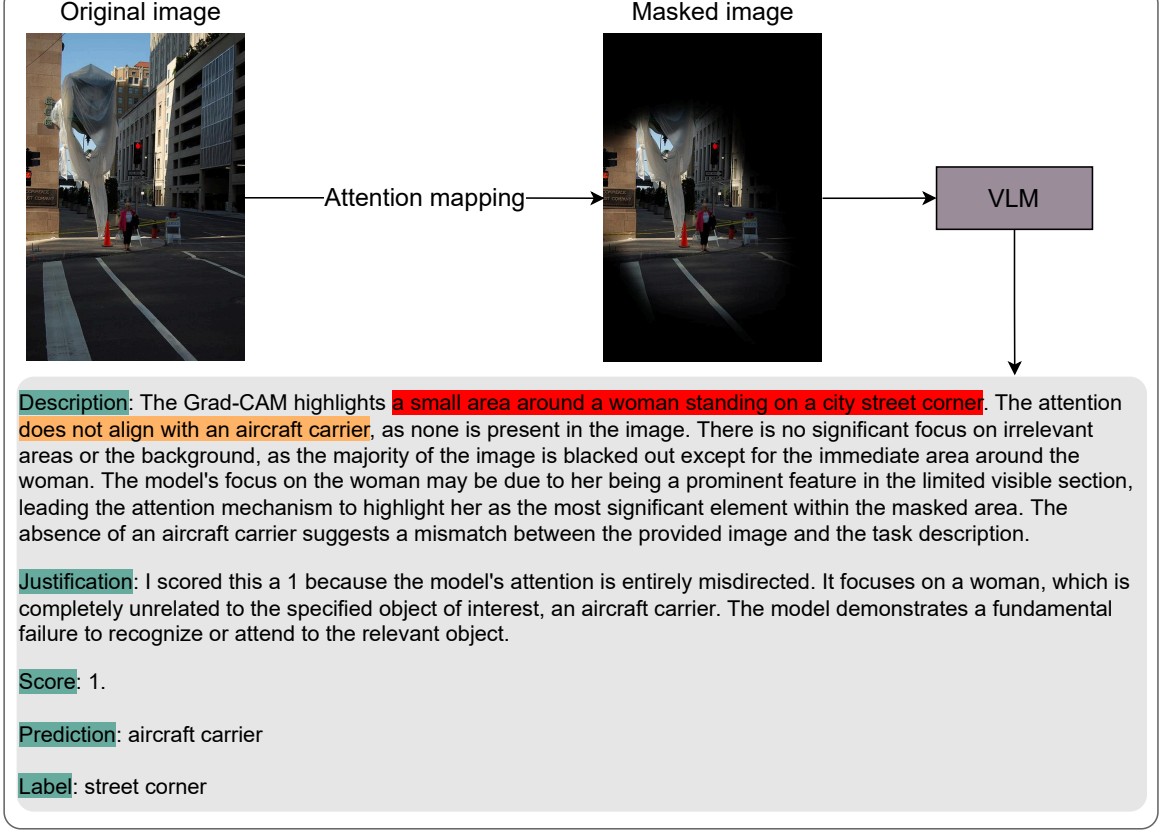

Figure 2: **Two prediction examples of the proposed pipeline.**

## A.5 Prompting

The prompt used for the evaluation framework consists of an image description, evaluation criteria, scoring, and output format. The task involves analyzing a masked image in which the model's focused areas are highlighted, while irrelevant regions are blacked out. Key criteria for evaluation include focus accuracy, object recognition, object coverage, and potential distractions from background or irrelevant elements. The evaluator is instructed to analyze the model's attention on the object and provide an explanatory analysis, considering factors like visual challenges or misleading elements. A score from 0 to 5 is assigned, with specific descriptions for each score reflecting the model's attention and recognition performance. The output includes a concise evaluation and score with justification.

---

**Prompt to get sample description justification and score from masked CAM images**

Task: Evaluate the Model's Attention Mechanism Using the Provided Masked Image.

- Image Description:
  - The image is masked with a Grad-CAM heatmap, where only the areas the model focuses on are visible, while all other regions are blacked out.
  - The model is attempting to focus on the object.

- Evaluation Criteria:
  - Focus Accuracy: Analyze which part of the image the Grad-CAM is highlighting. Is the model's attention placed accurately on the object, or is it scattered across other areas?
  - Object Recognition: Determine whether the model correctly recognizes the object. Is the attention primarily on the correct object, or does the model focus on irrelevant areas?
  - Object Coverage: Evaluate how much of the object is being captured by the model's attention. Is the entire object covered, only a small part, or none at all?
  - Background and Irrelevant Focus: Check for any significant focus on the background or irrelevant objects. Does this distract the model from the primary object?
  - Explanatory Analysis: Provide possible reasons for the model's attention pattern. Consider whether the model is being misled by similarly shaped or colored objects, complex backgrounds, or other visual challenges.

- Scoring:

  Assign a score between 0 and 5 based on the relevance and accuracy of the model's attention:
  - 0: The model's attention is completely irrelevant to the object, leading to a wrong result.
  - 1: The model fails to recognize the object entirely, focusing on irrelevant areas.
  - 2: The model captures only a small part of the object.
  - 3: The object is recognized, but the attention also covers irrelevant parts or other objects.
  - 4: Most of the object is detected correctly, with minimal distraction from irrelevant areas or the background.
  - 5: The model perfectly captures the entire object without being distracted by irrelevant areas or the background.

- Output Format:
  - Evaluation: Provide a concise evaluation (5-6 sentences), discussing: Where the Grad-CAM is focusing. Whether the attention aligns with the object. Whether there is any significant focus on irrelevant areas or the background. Explain why the model might focus on specific regions.
  - Score: Assign a score from 0 to 5, justifying your rating based on the model's performance in recognizing the object and avoiding distractions.

---

– The format must be presented as follows:

* Evaluation: [evaluation],
* Justification: [justification],
* Score: [score]

---

**Prompt to get sample description justification and score from original CAM images**

Task: Conduct an evaluation of the model's attention mechanism by analyzing its response to the supplied CAM heatmap. This assessment aims to test the model's capacity to effectively interpret and utilize attention when processing visual data.

- Image Description:

  – The heatmap uses warm colors (orange, red) to represent areas where the model is focusing most, while cool colors (blue, purple, dark) indicate regions of little to no attention.
  – The model's focus is on the object.
  – Identify the warm-colored regions and analyze what those regions represent in relation to the object of interest. In addition, assess the presence of cool-colored regions and their alignment with irrelevant areas or the background.

- Evaluation Criteria:

  – Focus Accuracy: Analyze which part of the heatmap the warm colors (orange, red) highlight. Is the model's attention accurately placed on the object, or is it scattered across other areas?
  – Object Recognition: Determine if the model is correctly recognizing the object. Is the attention primarily on the correct object, or does the model focus on irrelevant areas?
  – Object Coverage: Evaluate how much of the object is being captured by the model's attention. Is the entire object covered, only a small part, or none at all?
  – Background and Irrelevant Focus: Check for any significant focus on cool-colored regions. Does this distract the model from the primary object?
  – Explanatory Analysis: Provide possible reasons for the model's attention pattern. Consider whether the model is being misled by similar-colored areas, complex backgrounds, or other visual challenges.

- Scoring:

  Assign a score between 0 and 5 based on the relevance and accuracy of the model's attention:

  – 0: The model's attention is scattered with no clear target, showing that it does not understand the task or the object.
  – 1: The model consistently directs its attention to something unrelated to object, indicating a fundamental misunderstanding of the object it is supposed to recognize.
  – 2: Partial object recognition: The model captures only a small fragment of the object, missing most of its critical features. The attention is mostly misdirected, with just minor alignment to the actual object.
  – 3: The model identifies a limited area of object, but its attention still includes some irrelevant parts surrounding it.
  – 4: The model predominantly focuses on object, with only minor distractions or irrelevant attention in the background.
  – 5: The model accurately captures the entire object without any distractions from irrelevant areas or background elements.

- Output Format:
  - Evaluation: Provide a concise evaluation (5-6 sentences), discussing: Where the heatmap focuses (warm colors). Whether the attention aligns with the object. Whether there is any significant focus on irrelevant areas or the background. Explain why the model might be focusing on specific regions.
  - Score: Assign a score from 0 to 5, justifying your rating in a sentence.
  - Your output format must be presented in a dictionary as follows, which is extremely important for the evaluation process to run without any error:
    * Evaluation: [evaluation],
    * Justification: [justification],
    * Score: [score]

