# OpenReview forum: "Towards Scalable Explainable AI: Using Vision-Language Models to Interpret Vision Systems"
_TMLR — Rejected by TMLR_

### Review · Reviewer_mDnM · 2026-02-25

**Summary Of Contributions:**

This paper introduces a pipeline designed as a scalable explanation framework for image-based models. It combines a “CAM-based XAI method” with a vision-language model that automates the evaluation of explanations. By aggregating scores produced by the vision-language model across multiple samples, the authors summarize the model's behavior in the form of a confusion matrix.
To assess the viability of their approach, they compare the scores obtained against human assessments and other established metrics. The pipeline is then applied to the downstream task like detecting defective samples in datasets.

Strengths

- The proposed approach is straightforward and easy to follow. The full pipeline is clearly explained, including technical details; notably, hyperparameter studies and prompts are provided in the appendix.
- The application section is a welcome addition, and the results presented there are interesting.
- To the best of my knowledge, the perspective of evaluating models through perceptual interpretation using vision-language models is novel.

Weaknesses

- There is, in my view, a lack of awareness of the state of the art. The authors claim that "the applications to analyze interpretive visualizations, such as Grad-CAM, in visual models remain underexplored" — a statement I do not believe to be accurate. The field of neural network explanation evaluation offers a wide range of relevant tools [1, 2], including work specifically addressing the perceptual aspect of explanations, i.e., the extent to which explanations are interpretable by humans [3, 4]. Potentially relevant related works also include those producing global explanations [5, 6].
- The focus on the perceptual aspect should be accompanied by a discussion of its inherent limitations. Is having a model that behaves like a human truly the ultimate goal? Furthermore, there exists substantial criticism regarding the faithfulness of XAI methods to the underlying model [8] — it remains unclear how the proposed pipeline ensures that it captures the model's true behavior.
- There is no discussion of computation time or costs, which is particularly important given that the authors emphasize scalability as a key advantage of their approach.
- The experiments assessing the viability of the pipeline feel limited: the evaluation essentially reduces to a comparison of scores with human assessments, with little information provided about how those human scores were collected. Moreover, parts of the process — such as the justification step, which appears to be integral to the pipeline — are not validated.
- Some of the contributions presented by the authors are debatable. For instance, the contribution called Masked-CAM — which displays the explanation as the input image masked by the activation map — is a visualization option already available in existing libraries [11]. Similarly, presenting results as a confusion matrix is a questionable contribution in itself.
- The term "CAM-based" is used loosely throughout the paper, appearing to refer broadly to any method with "CAM" in its name. See the requested changes section for specific suggestions.
- Relatedly, the method used to produce attention maps is unclear. The authors state that "different methods to extract models' attention, including CAM, LayerCAM, and more, are utilized," but it is not specified whether the resulting activation map corresponds to a single method or a combination thereof, nor which methods are concretely used in the experiments. Furthermore, the exclusive focus on CAM-based methods is questionable, as several other approaches — such as LIME, SHAP, RISE, Deep Feature Factorization, or BCos — can also produce image masks and would seemingly be compatible with the proposed pipeline.

**Additional Comments:**

References:

[1] Hedström, A., Weber, L., Krakowczyk, D., Bareeva, D., Motzkus, F., Samek, W., ... & Höhne, M. M. C. (2023). Quantus: An explainable ai toolkit for responsible evaluation of neural network explanations and beyond. Journal of Machine Learning Research, 24(34), 1-11.

[2] Nauta, M., Trienes, J., Pathak, S., Nguyen, E., Peters, M., Schmitt, Y., ... & Seifert, C. (2023). From anecdotal evidence to quantitative evaluation methods: A systematic review on evaluating explainable ai. ACM Computing Surveys, 55(13s), 1-42.

[3] Kazmierczak, R., Azzolin, S., Berthier, E., Hedström, A., Delhomme, P., Filliat, D., ... & Franchi, G. (2024). Benchmarking xai explanations with human-aligned evaluations. AAAI 2026

[4] De Bona, F. B., Dominici, G., Miller, T., Langheinrich, M., & Gjoreski, M. (2024). Evaluating explanations through llms: Beyond traditional user studies. arXiv preprint arXiv:2410.17781.

[5] Shaham, T. R., Schwettmann, S., Wang, F., Rajaram, A., Hernandez, E., Andreas, J., & Torralba, A. (2024, July). A multimodal automated interpretability agent. In Forty-first International Conference on Machine Learning.

[6] Parola, M., Alfeo, A. L., & Cimino, M. G. (2026). Human-centered XAI via a Concept-Informed Prompt-based Validation framework for saliency maps [CIProVa]. Image and Vision Computing, 105920.

[8] Kindermans, P. J., Hooker, S., Adebayo, J., Alber, M., Schütt, K. T., Dähne, S., ... & Kim, B. (2019). The (un) reliability of saliency methods. In Explainable AI: Interpreting, explaining and visualizing deep learning (pp. 267-280). Cham: Springer International Publishing.

[9] Colin, J.; Fel, T.; Cadene, R.; and Serre, T. 2022. What I ` cannot predict, I do not understand: A human-centered evaluation framework for explainability methods. Advances in Neural Information Processing Systems, 35: 2832–2845.

[10] Morrison, K.; Jain, M.; Hammer, J.; and Perer, A. 2023. Eye into AI: Evaluating the interpretability of explainable AI techniques through a game with a purpose. Proceedings of the ACM on Human-Computer Interaction, 7(CSCW2): 1–22.

[11] https://github.com/jacobgil/pytorch-grad-cam

**Audience:**

Yes

**Audience Explanation:**

The perceptual assessment of models is a valuable direction in the context of growing interest in AI trustworthiness. The proposed pipeline is readily applicable in an automated fashion, which is particularly appealing given that such evaluations currently rely on tedious human labor.

**Claims And Evidence:**

No

**Claims Explanation:**

The proposed pipeline lacks sufficient empirical validation. Several architectural choices are inadequately justified — most notably the selection of the XAI method — and the experiments remain limited: the textual component receives no validation, and the scoring part is assessed on a single dataset with a single experiment. Finally, the paper insufficiently situates itself within the existing literature, with only one related work cited, which risks misleading the reader about the true novelty of the contribution.

**Requested Changes:**

Major:

- Better situate the paper within the existing landscape. Provide a clearer comparison with, or argumentation against, related works on XAI evaluation methods and global XAI approaches.
- Report computation times and costs.
- Add experiments to strengthen the validation of the pipeline. On the scoring side, compare against existing human-computer interaction datasets such as [3, 9, 10]. On the textual side, consider using captioning benchmarks as an evaluation framework.
Reorganize the contributions section. The 2nd and 3rd stated contributions are problematic as currently framed. Conversely, the application section — which I consider a genuine contribution — is not highlighted as such and should be.
- Describe more precisely how activation maps are computed within the pipeline, and justify this choice either through an ablation study or through principled arguments.
- Provide more details about how human ratings were collected: for example, how many annotators were involved, what they were asked to do, and whether they had any background knowledge in the field.

Minor:
- Broaden the limitations section to address the implications of evaluating models exclusively through the perceptual lens.
Reconsider the term "CAM-based methods". Possible replacements include saliency-based methods (methods that produce explanations in the form of a saliency map) or (image) attribution-based methods (methods that output explanations as a scalar value assigned to each input position, i.e., pixels in the case of images).
- Where relevant, direct the reader explicitly to the corresponding appendix sections. As a concrete example, Section 3.2 should mention that the prompts used are available in Appendix A.2.

---

> ### Author Response · Authors · 2026-03-14
>
> We thank the reviewer for the constructive feedback. We will address the concerns as follows:
>
> **Related Work and Novelty:** We will expand the related work section to include comprehensive comparisons with existing XAI evaluation methods [1-6], perceptual explanation studies [3,4], and global XAI approaches [5,6]. We clarify that our contribution lies not in proposing novel evaluation metrics, but in automating perceptual assessment through vision-language models, which enables scalable, self-evaluating model analysis without human labor.
>
> **Computational Cost:** We will report detailed computational costs in the paper. For our implementation using Gemini models, processing will cost approximately $0.00016 per sample (563 input tokens + 258 image tokens -> ~200 output tokens). Runtime is about 2.5 seconds (depending on the network) per sample via API call on any hardware. Our implementation does not apply batching for speed or cost optimization.
>
> **Validation and Experiments:** We will strengthen validation through comparison with HCI datasets [3,9,10] for scoring evaluation. However, to evaluate the description, our pipeline uses masked images to generate descriptions, while most benchmarks focus on full image captioning. For that reason, we can either evaluate using a human annotator (this is what we did in our paper) or use the VLM-as-a-judge pipeline on a small dataset.
>
> **Justification Evaluation:** Currently, the generated description, justification, and score are evaluated at the same time in experiment 4.1.2, not individually. This is because we consider the justification step an intermediate reasoning step that generates a score more consistent with our defined criteria in the prompt, which might not need to be evaluated individually, but only through its consistency with the description and score from the pipeline.
>
> **Contributions:** We will reorganize the contributions section. Masked-CAM will be removed as it exists in prior libraries [11]. The confusion matrix will be reframed as a deliberate design choice (not a contribution) that enables quantitative alignment checking with human raters and automated validation through segmentation metrics, which is essential for a self-evaluating system. The application section will be highlighted as a primary contribution.
>
> **Human Evaluation Details:** Two annotators with an AI background independently score each sample and evaluate both generated texts from VLM (description, justification, score). While we acknowledge that this can be a limitation to our work, we provide different evaluation metrics like IOU, PA, Dice, and F1 in our evaluation.
>
> **References:**
> [1] Hedström, A., Weber, L., Krakowczyk, D., Bareeva, D., Motzkus, F., Samek, W., ... & Höhne, M. M. C. (2023). Quantus: An explainable ai toolkit for responsible evaluation of neural network explanations and beyond. Journal of Machine Learning Research, 24(34), 1-11.
> [2] Nauta, M., Trienes, J., Pathak, S., Nguyen, E., Peters, M., Schmitt, Y., ... & Seifert, C. (2023). From anecdotal evidence to quantitative evaluation methods: A systematic review on evaluating explainable ai. ACM Computing Surveys, 55(13s), 1-42.
> [3] Kazmierczak, R., Azzolin, S., Berthier, E., Hedström, A., Delhomme, P., Filliat, D., ... & Franchi, G. (2024). Benchmarking xai explanations with human-aligned evaluations. AAAI 2026
> [4] De Bona, F. B., Dominici, G., Miller, T., Langheinrich, M., & Gjoreski, M. (2024). Evaluating explanations through llms: Beyond traditional user studies. arXiv preprint arXiv:2410.17781.
> [5] Shaham, T. R., Schwettmann, S., Wang, F., Rajaram, A., Hernandez, E., Andreas, J., & Torralba, A. (2024, July). A multimodal automated interpretability agent. In Forty-first International Conference on Machine Learning.
> [6] Parola, M., Alfeo, A. L., & Cimino, M. G. (2026). Human-centered XAI via a Concept-Informed Prompt-based Validation framework for saliency maps [CIProVa]. Image and Vision Computing, 105920.
> [8] Kindermans, P. J., Hooker, S., Adebayo, J., Alber, M., Schütt, K. T., Dähne, S., ... & Kim, B. (2019). The (un) reliability of saliency methods. In Explainable AI: Interpreting, explaining and visualizing deep learning (pp. 267-280). Cham: Springer International Publishing.
> [9] Colin, J.; Fel, T.; Cadene, R.; and Serre, T. 2022. What I ` cannot predict, I do not understand: A human-centered evaluation framework for explainability methods. Advances in Neural Information Processing Systems, 35: 2832–2845.
> [10] Morrison, K.; Jain, M.; Hammer, J.; and Perer, A. 2023. Eye into AI: Evaluating the interpretability of explainable AI techniques through a game with a purpose. Proceedings of the ACM on Human-Computer Interaction, 7(CSCW2): 1–22.
> [11] https://github.com/jacobgil/pytorch-grad-cam

---

### Review · Reviewer_DgmD · 2026-03-07

**Summary Of Contributions:**

This paper proposes an automated evaluation pipeline that leverages vision-language models (VLMs) to analyze the behavior of vision models through saliency maps and generate interpretive scores and summaries.

However, the technical contributions of the paper appear limited. The primary methodological component is the introduction of a shifted sigmoid function in Equation (1) to transform attention masks when generating masked images. While this modification may help adjust the visual emphasis of saliency maps, it represents only a minor adjustment and does not constitute a substantial algorithmic contribution to explainable AI or model evaluation.

More broadly, the paper mainly integrates existing components—including saliency methods and large vision-language models—without introducing new techniques for attribution, explanation generation, or faithfulness evaluation. As a result, the overall novelty of the proposed pipeline appears limited, with the main contribution lying more in system integration than in methodological advancement.

**Additional Comments:**

The relationship between this work and the LangXAI framework (Nguyen et al., 2024) is also not entirely clear. The proposed pipeline appears to follow a similar structure—combining saliency maps with vision-language models to generate interpretations—but extends the process toward scoring and dataset-level analysis. However, the distinction between the proposed method and LangXAI is not sufficiently articulated. In particular, it remains unclear whether this work introduces a fundamentally new framework for automated evaluation or largely modifies the LangXAI pipeline by adding aggregation and scoring components. From the reviewer’s perspective, the proposed approach appears to be a relatively minor extension of the LangXAI framework rather than a substantially new methodological contribution.

The choice of the hyperparameters $\alpha$ and $\beta$ also raises several concerns. First, the ablation study appears to be conducted on only a single dataset, which limits the generalizability of the conclusions. Evaluating the sensitivity of these parameters on a single dataset provides insufficient evidence that the proposed transformation is robust across different datasets, model architectures, or vision tasks.

Second, the reported parameter settings ($\alpha = 25, \beta = 0.4$; $\alpha = 15, \beta = 0.6$; $\alpha = 25, \beta = 0.7$) appear somewhat selective. It is unclear how these particular combinations were chosen, and they seem to be presented primarily because they outperform the baseline CAM case. Without a systematic exploration of the parameter space or a comprehensive sensitivity analysis, this raises the concern that the reported improvements may result from **cherry-picking favorable hyperparameter configurations** rather than reflecting a consistently effective method.

More broadly, the paper does not clarify whether other parameter combinations were tested and how they performed. A more convincing evaluation would include a systematic grid search or sensitivity analysis over a wider range of $\alpha$ and $\beta$ values, ideally across multiple datasets, to demonstrate that the observed improvements are stable and not dependent on a small number of carefully selected configurations.

**Audience:**

Yes

**Audience Explanation:**

CAM-based XAI techniques play an important role in understanding and interpreting the behavior of deep learning models.

**Broader Impact Concerns:**

The evaluation pipeline is not described with sufficient clarity, which may hinder reproducibility and make it difficult for future work to replicate the reported results.

**Claims And Evidence:**

No

**Claims Explanation:**

The evaluation relies on human ratings from only two annotators, both of whom are authors of the paper. Although the reported Pearson correlation between the two raters is 0.71, this level of agreement is relatively low and indicates only moderate consistency, raising concerns about the stability and objectivity of the annotations. Moreover, using the paper’s authors as annotators introduces potential bias in the evaluation process.

The paper also states that *“to avoid multi-object predictions, we filter for samples in which a single (or main) object occupies more than one-third of the image area and falls within the ResNet18 output classes, ensuring a dominant subject for analysis.”* However, the filtering procedure is not described in sufficient detail, and no statistics are provided regarding how many samples were removed or how the remaining dataset distribution was affected. This lack of transparency makes the evaluation pipeline difficult to assess and potentially non-reproducible.

Furthermore, restricting the dataset to images containing a single dominant object occupying more than one-third of the image area significantly simplifies the task. While this may facilitate analysis, it removes many realistic scenarios such as multi-object scenes, cluttered environments, and small objects. Consequently, the evaluation may overestimate the method’s performance and does not convincingly demonstrate its effectiveness in more complex and realistic settings.

A more reliable evaluation would involve multiple independent annotators, a clearly defined annotation protocol, and the reporting of standard inter-rater reliability metrics to ensure the robustness and objectivity of the human judgments.

**Requested Changes:**

Equation (1) could be presented more clearly using a standard sigmoid formulation. For example, rewriting it as


$\sigma(\alpha (v - \beta))$

would make the roles of the scaling parameter $\alpha$ and the shift parameter $\beta$ more explicit and easier to interpret.

Please also add more ablation studies regarding these hyperparameters with different values on different datasets.

And more visualization on representative samples.

---

> ### Author Response · Authors · 2026-03-14
>
> We thank the reviewer for the detailed feedback. We will address the concerns as follows:
>
> **Contribution and Novelty:** Our core contribution lies in the **automated aggregation framework** that enables scalable, self-evaluating model analysis. The shifted sigmoid in Equation (1) is a design choice (not a core algorithmic contribution) to ensure masked images maintain sufficient visual variance for reliable VLM assessment, preventing degenerate cases where masks are either completely black or indistinguishable. We will clarify that our novelty is in addressing the critical gap of manual per-sample inspection using VLM, not in proposing new attribution techniques. While both pipelines (ours and LangXAI) generate descriptions from saliency maps, LangXAI focuses on **individual sample explanation** requiring manual inspection. Our pipeline addresses the critical need for **automated aggregation and evaluation at scale**, which is essential as AI-generated data volumes grow. This distinction represents a fundamentally different application scenario, not a minor extension.
>
> **Human Evaluation and Bias:** We acknowledge that using two author-annotators with moderate agreement (Pearson r=0.71) is a limitation. To mitigate potential bias, we employ automated segmentation metrics (IOU, Dice, F1, Pixel Accuracy) as an alternative evaluation. These metrics show consistent rankings with our method, confirming that results are not driven by annotator bias.
>
> **Dataset Filtering:** Initially, our idea is to convert the COCO dataset into ImageNet, a classification dataset in which each image contains only one object (which is usually the most confident prediction of the vision model), which is why we use filtering to create the data. We will run on different benchmarks like [1,2,3] to enhance the evaluation or remove the filtering in this experiment.
>
> **Equation (1) and Hyperparameters:** Parameter selection was guided by two criteria: (i) masked images must retain visible differences to the naked eye (ensuring VLM assessment responds to actual visual changes, not noise), and (ii) masks must avoid complete black-out or no masking extremes. We will try with more settings in this experiment. The hyperparameter settings were chosen randomly based on these criteria.
>
> **References**
> [1] Kazmierczak, R., Azzolin, S., Berthier, E., Hedström, A., Delhomme, P., Filliat, D., ... & Franchi, G. (2024). Benchmarking xai explanations with human-aligned evaluations. AAAI 2026
> [2] Colin, J.; Fel, T.; Cadene, R.; and Serre, T. 2022. What I cannot predict, I do not understand: A human-centered evaluation framework for explainability methods. Advances in Neural Information Processing Systems, 35: 2832–2845.
> [3] Morrison, K.; Jain, M.; Hammer, J.; and Perer, A. 2023. Eye into AI: Evaluating the interpretability of explainable AI techniques through a game with a purpose. Proceedings of the ACM on Human-Computer Interaction, 7(CSCW2): 1–22.

---

### Review · Reviewer_tdJ6 · 2026-03-09

**Summary Of Contributions:**

This paper proposes a pipeline that leverages VLMs to score and summarize CAM‑based explanations at dataset scale, and shows that the resulting scores enable automated, confusion‑matrix–style analysis of how different vision architectures use attention, and can be used as a practical tool for detecting flawed or noisy samples in large vision datasets, with performance competitive with specialized label‑noise baselines.

Strengths: The method is simple and modular. It is easy to reproduce and make the usage practical. The method also perform competitively among result methods.

Weaknesses see requested changes

**Audience:**

Yes

**Audience Explanation:**

It is an existence proof that VLM-as-a-judge on masked attention regions yields better-behaved and more interpretable metrics than classic perturbation scores. Also those metrics are informative enough to drive concrete downstream tasks like model comparison and data cleaning.

**Broader Impact Concerns:**

/

**Claims And Evidence:**

Yes

**Claims Explanation:**

Yes, the results look convincing and also honest limitations and scope. They explicitly discuss reliance on VLM quality and restriction to CAM-based xAI, and position the method as a support tool rather than an automatic oracle.

**Requested Changes:**

C1) The abstract feels poorly written. It reads more like an intro paragraph than a tight abstract. Please revise it and also mention the experiments and results. Try 1-2 sentence on motivation, 1-2 sentence on design, 1-2 sentence on experiment and results (mention quantitive numbers).

C2) The citation style also makes it uneasy to read. try to format this way Chu et al. (2024) -> (Chu et al. , 2024).

C3) Missing full term on CAM, where Class Activation Mapping should have mentioned once before using the aberration.

C4) The reliability of VLM experiment feels limited. Is it possible to report the correlation on human vs VLM rated on attention 0-5 scores? A subset would be sufficient.

---

> ### Author Response · Authors · 2026-03-14
>
> We thank the reviewer for the positive assessment and constructive suggestions. We will address the requested changes as follows:
>
> **C1) Abstract Revision:** We will rewrite the abstract to follow a tighter structure following your recommendations.
>
> **C2) Citation Style:** We will reformat all citations from "Chu et al. (2024)" to "(Chu et al., 2024)" throughout the paper for consistency and readability.
>
> **C3) CAM Terminology:** We will introduce the full term "Class Activation Mapping (CAM)" at first usage, with the abbreviation defined, before using "CAM" exclusively in subsequent mentions.
>
> **C4) VLM Reliability Validation:** The correlation between human and different VLMs is reported in Table 1 (Human metrics).

---

### Comment · Action_Editor_iuQg · 2026-03-15

Dear reviewers,

the authors have now responded to your reviews. I would hence like to ask you to consider these carefully and engage with the authors in a discussion. In this, please do not focus only on your own review and responses but check also the other parts of the discussion.
There seem to be some spread in the reviews and opinions - have other reviewers brought up important points that you may have missed? Or there some points in the reviews that you find not so critical?

In the responses the authors proposed multiple updates to the papers - are these sufficient to address your concerns?
Do you believe that these can be integrated into the paper in a simple update? Or do you feel a more substantial revision of the paper would be beneficial - if so, in what direction?

Thanks,
your AE

---

> ### Comment · Reviewer_mDnM · 2026-03-16
> **Concerns about the rebutal**
>
> In my view, the response is somewhat incomplete. While I do not expect all requested changes to be addressed during the review process, the authors' reply consists largely of promises of future modifications — which, in my opinion, warrants at least partial implementation in the manuscript.
> That said, I believe this may stem from a misunderstanding of the TMLR process on the authors' part. As a reminder, it is possible to submit a revised version of the manuscript during the reviewing process. A common practice is to highlight changes in a different color, allowing reviewers to easily identify what has been modified. This interim revision is distinct from the final revision that would be requested should the paper receive an "Accept with Minor Revisions" decision.

---

### Author Response · Authors · 2026-04-14
**Inquiry on Manuscript Status**

Dear Editors,

I hope you are doing well.

I am writing to kindly inquire about the current status of our manuscript.

We have carefully addressed all reviewers' comments and submitted a revised version along with detailed responses. Since it has been a while since our last update, I would greatly appreciate any information you could share regarding the progress of the evaluation, as well as any further feedback from the reviewers or action editors.

We fully understand that the review process can take time, and we sincerely appreciate the effort and coordination involved. I just wanted to check in to ensure that no additional information is required from our side.

Thank you very much for your time and consideration.

Best regards,
Authors

---

### Decision · Action_Editor_iuQg · 2026-04-16

**Recommendation:** Reject

**Additional Comments:**

The reviewers have raised multiple concerns that haven't been sufficiently addressed. These relate mainly to the experimental setup ( objectivity of evaluation, reproducibility, selection of baseline) and the overall positioning within the existing landscape in the field. Overall, the authors are advised to consider the reviews and the questions, concerns and recommendations therein more carefully, before submitting an improved version.

**Audience:**

Yes

**Audience Explanation:**

AI explainability and trustworthiness is an increasingly important topic and solid contribution to the state of the art in this direction would without any doubts attract attention of the community.

**Claims And Evidence:**

No

**Claims Explanation:**

In the review process the reviewers have raised multiple significant concerns. The authors addressed these partially in the discussion and have introduced some improvements into the paper, yet the reviewers haven't found these to be sufficient for addressing all the raised issues.

**Resubmission Of Major Revision:**

The authors may consider submitting a major revision at a later time.